# Postoperative Clinical Outcomes of Thoracoscopic Surgery under Local Anesthesia in Patients with Primary Spontaneous Pneumothorax

Eunji Kim [1], Chi-Seung Lee [2,*], Jeong Su Cho [3], Hoseok I [3], Yeong Dae Kim [3], Eunsoo Kim [4] and Hyo Yeong Ahn [3,*]

[1] Department of Thoracic and Cardiovascular Surgery, Asan Medical Center, University of Ulsan College of Medicine, Seoul 05505, Korea; kej2683@gmail.com
[2] Department of Convergence Medicine and Biomedical Engineering, School of Medicine, Pusan National University, and Biomedical Research Institute, Pusan National University Hospital, Busan 49241, Korea
[3] Department of Thoracic and Cardiovascular Surgery and Biomedical Research Institute, Pusan National University Hospital, Busan 49241, Korea; drmozart@hanmail.net (J.S.C.); ihoseok@pusan.ac.kr (H.I.); domini@pnu.edu (Y.D.K.)
[4] Department of Anesthesia and Pain Medicine and Biomedical Research Institute, Pusan National University Hospital, Busan 49241, Korea; eunsookim@pusan.ac.kr
* Correspondence: victorich@pusan.ac.kr (C.-S.L.); doctorahn02@hanmail.net (H.Y.A.)

**Abstract:** (1) Background: since the technologies of anesthesia and surgery were advanced, video-assisted thoracic surgery (VATS) under local anesthesia (LA) has been widely carried out and is considered a robust surgical technique to prevent the recurrence of pneumothorax in patients with recurrent primary spontaneous pneumothorax (PSP). In this study, postoperative clinical outcomes were compared to evaluate the feasibility and efficacy of VATS under LA compared with general anesthesia (GA) in patients with PSP. (2) Methods: 255 patients underwent wedge resection underwent VATS for PSP in our hospital from January 2014 to June 2019. Of them, 30 patients underwent the operation under LA and the others underwent the operation under GA. Except for the anesthesia method, the same surgical technique was adopted for all patients. All medical records were retrospectively reviewed. (3) Results: the total operation time and total hospital days were relatively shorter, post-chest tube drainage was significantly shorter (0.04), and visual analog scale (VAS) scores in the outpatient clinic were significantly lower in the LA group than in the GA group ($p = 0.01$). The incidence of postoperative recurrence after discharge in the LA group (3.3%) was also lower than in the GA group (18.67%) ($p = 0.001$). In the LA group, there were no cases of conversion to intubation. (4) Conclusions: our results showed relatively better clinical outcomes in VATS under LA with sedation than under GA in the treatment of PSP. Hence, LA with sedation can be considered as a robust anesthetic technique for VATS and as applicable in the surgical treatment of PSP.

**Keywords:** primary spontaneous pneumothorax (PSP); local anesthesia (LA); general anesthesia (GA); clinical outcome

## 1. Introduction

Spontaneous pneumothorax is a commonly encountered disease. If there is no underlying lung disease, it is categorized as primary spontaneous pneumothorax (PSP). Although there is no underlying pulmonary disease in PSP patients, blebs and bullae are observed in computed tomography (CT) imaging in up to 80–90% of the cases [1]. PSP is widely known to occur in tall, thin young men and shows a high recurrence rate in patients who undergo no surgical interventions [2–6]. Video-assisted thoracic surgery (VATS) can be performed to prevent recurrences after closed thoracostomy procedures.

VATS is usually performed under general anesthesia (GA) with double-lumen endotracheal intubation. However, general anesthesia may be associated with complications

including lung infections, intubation-related airway trauma, injury due to ventilation pressure or overexpansion, residual neuromuscular blockade, cardiac dysfunction, arrhythmia, postoperative sore throat, and postoperative nausea and vomiting [7–10].

For these reasons, our center conducts surgery for PSP under non-intubated anesthesia, specifically local anesthesia (LA) with sedation. This study aimed to prove the feasibility and efficacy of VATS performed under LA with sedation in PSP patients.

## 2. Materials and Methods

From January 2014 to June 2019, 255 patients underwent wedge resection using VATS for PSP in our hospital. The operation indications in these patients included recurrent ipsilateral or contralateral pneumothorax, persistent air leakage for more than seven days, total collapse in initial X-ray scans, large or multiple visible blebs in chest CT images, concomitant hemothorax, and frequent pneumothorax recurrence due to stress, caused by upcoming important exams such as the Korean Collage Scholastic Ability Test (CSAT) among Korean student patients. Indeed, according to some literature, this extreme stress can lead to the frequent recurrence of pneumothorax (Table 1).

**Table 1.** Prevalence of indications in each group.

| Variable | GA Group (n = 225) | LA Group (n = 30) | *p*-Value |
|---|---|---|---|
| Recurrent ipsilateral pneumothorax | 101 | 8 | 0.06 |
| Contralateral pneumothorax | 21 | 3 | 0.91 |
| Persistent air leakage for more than seven days | 15 | 1 | 0.48 |
| Total collapse in initial x-ray scans | 20 | 5 | 0.18 |
| Large or multiple visible blebs in chest CT images | 26 | 2 | 0.42 |
| Concomitant hemothorax | 2 | 1 | 0.24 |
| Upcoming important examinations | 40 | 10 | 0.11 |

Of these 255 patients, 30 underwent surgery under LA, while the others underwent surgery under GA. The surgical technique was more or less identical in both groups; however, three air-locking ports were used in LA in case of abrupt desaturation or if endurable dyspnea was caused.

All medical records were retrospectively reviewed for operating time, interval time to operation, duration of pre- and post-procedural chest tube drainage, total duration of hospital stay, visual analog scale (VAS) scores measured at various points in time (preoperatively, immediately after surgery, on postoperative day (POD) 1, and at the first outpatient visit during follow-up), post-procedural complications, duration of post-procedural hospital stay, number of wedge resections and number of ports inserted, technique used for pleural coverage, duration of postoperative air leak, recurrence rate, and follow-up duration. This study was approved by the Institutional Review Board of Pusan National University, Busan, Republic of Korea (IRB No. 1908020084). Informed consent was not required for this retrospective study.

### 2.1. Anesthetic Techniques

2.1.1. Local Anesthesia with Sedation (LA Group)

Precedex™ (2.5 mg/kg) and fentanyl (2 µg/kg) were injected through an intravenous line until the patient was sedated; administration was stopped just after wedge resection was completed. Oxygen was supplied at a rate of 3 L/min through the nasal cannula. Standard monitors were in place including a five-lead ECG (electrocardiogram), non-invasive arterial blood pressure monitoring, and $SpO_2$ measurement.

2.1.2. General Anesthesia (GA Group)

For patients in this group, propofol (2.5 mg/kg) and fentanyl (2 µg/kg) were administered intravenously; intubation was done with a cuffed endotracheal tube after muscle

relaxation was achieved using cisatracurium (0.2 mg/kg). Following the initial dose, anesthesia was maintained using an isoflurane and cisatracurium maintenance dose (0.2 mg/kg every 50–60 min). Positive pressure ventilation was used to obtain an end-tidal $CO^2$ level of 35 mmHg, at which point an oropharyngeal pack was inserted.

## 2.2. Operative Techniques

Patients were placed in the lateral position with the operating side facing upwards; 2% lidocaine was then injected into the previous chest tube site. While thoracoscopic inspection was performed, 2% lidocaine was injected into the following sites, according to the technique used; a 5 mm port in the sub-axillary line, a 2 mm port in the fifth prescapular line, and the site of the previous chest tube. After the thoracoscope was placed through the 5 mm port in the sub-axillary line, the blebs were grabbed using a 2 mm clinch and resected using a stapler inserted through the previous chest tube site.

After resection of the blebs, an air leak test was performed by applying suction through the air-locking port under a closed system in the LA group, and by maintaining ambu-bagging with two-lung ventilation in the GA group. The staple line was covered using non-woven polyglycolic acid sheets (Neoveil®(Gunze Limited; Ayabe, Japan)) and sealed with a fibrin sealant (Tisseel (Baxter Healthcare Corporation; Westlake Village, California, USA)) or oxidized regenerated cellulose with fibrin sealant (Surgicel®(Ethicon Inc.—Johnson & Johnson; Somerville, NJ USA)).

## 2.3. Postoperative Care

Generally, Jackson-Pratt (J-P) drains maintain a flat and round shape, allowing no remnant air leakage. However, in cases where there was bulging of the bag two or more times, the bag needed to be replaced with a bottle. Suction was applied until the air leak stopped. When x-rays showed full expansion of the lungs and complete cessation of the air leak, the J-P drain could be removed.

## 2.4. Statistical Analysis

Data were analyzed using the Statistical Package for Social Sciences®software (SPSS®) version 16.0 (IBM®; Chicago, Illinois, USA) and statistical significance was defined as $p < 0.05$. For univariate statistical analysis, the Mann—Whitney U test for clinical findings and the chi-squared test for gender distribution were used. Spearman's correlation coefficient was adopted to measure the strength association between the rate of recurrence and the number of ports inserted, the length of lung resected, the technique used for pleural coverage, the number of wedge resections performed, or the number of previous attacks.

## 3. Results

Of 255 patients, 225 underwent surgery under GA with endotracheal intubation while 30 underwent surgery under LA with sedation and without endotracheal intubation. There were no large differences between the demographic characteristics of the two groups (Table 2).

The mean age was $21.05 \pm 6.13$ years in the GA group and $21.76 \pm 5.71$ years in the LA group ($p = 0.98$). The number of male patients in the GA group was 210 and in the LA group 29 ($p = 0.15$) (Table 1). The mean interval from the day of admission to the operation, the mean interval from the operation to the time of discharge, and the total duration of hospital stay were significantly shorter in the LA group than in the GA group ($p = 0.03$, 0.08, <0.01). The durations of preoperative chest tube drainage were $2.96 \pm 2.31$ days in the GA group and $2.20 \pm 1.86$ days in the LA group ($p = 0.10$) and those of postoperative chest tube drainage were $1.58 \pm 1.27$ days in the GA group and $1.27 \pm 0.52$ days in the LA group ($p = 0.30$) The pain which patients complained of was measured using VAS scores. Although there were no significant differences between the two groups in preoperative VAS scores, on POD 1 ($p = 0.58$) the VAS scores measured in the outpatient clinic (during follow-up) were $0.15 \pm 0.62$ in the GA group and $0.00 \pm 0.00$ in the LA group ($p = 0.05$) (Table 3).

**Table 2.** Patient demographic characteristics.

| Variable | GA Group | LA Group | *p*-Value |
|---|---|---|---|
| | (n = 225) | (n = 30) | |
| Age (years) | 21.05 ± 6.13 | 21.76 ± 5.71 | 0.98 |
| Gender | | | 0.15 |
| Female | 15 | 1 | |
| Male | 210 | 29 | |
| Height (cm)/weight (kg) | | | 0.61 |
| Female | 164.71 ± 7.28/48.64 ± 8.34 | 167/47.6 | |
| Male | 185.56 ± 6.11/61.13 ± 9.64 | 184.94 ± 7.33/57.64 ± 6.49 | |
| Smoking history | | | 0.03 |
| Ex-smoker | 9 | 2 | 1.0 |
| Recent smoker | 9 | 4 | 0.82 |

**Table 3.** Perioperative clinical findings.

| Variable | GA Group | LA Group | *p*-Value |
|---|---|---|---|
| | (n = 225) | (n = 30) | |
| Interval day (days) | | | |
| Admission to operation | 3.42 ± 1.98 | 2.57 ± 1.61 | 0.03 |
| Operation to discharge | 2.56 ± 1.51 | 1.73 ± 1.23 | 0.08 |
| Total hospital stay (days) | 5.92 ± 2.55 | 4.33 ± 1.99 | <0.01 |
| Chest tube drainage | | | |
| Preoperative chest tube drainage (days) | 2.96 ± 2.31 | 2.20 ± 1.86 | 0.10 |
| Postoperative chest tube drainage (days) | 1.58 ± 1.27 | 1.27 ± 0.52 | 0.30 |
| Oxygen saturation postoperation (%) | 97.26 ± 1.74 | 97.97 ± 1.33 | 0.05 |
| VAS score | | | |
| Preoperative | 1.95 ± 1.25 | 1.97 ± 0.85 | 0.81 |
| Immediate | 5.07 ± 1.60 | 5.40 ± 1.16 | 0.44 |
| Postoperative day 1 | 2.80 ± 0.61 | 2.77 ± 0.50 | 0.58 |
| Outpatient clinics | 0.15 ± 0.62 | 0.00 ± 0.00 | 0.05 |
| Follow-up duration (days) | 314.25 ± 442.19 | 109.57 ± 178.69 | <0.01 |
| Recurrence | 42 (18.67%) | 1 (3.33%) | 0.03 |

The duration of the follow-up was 314.25 ± 442.19 days in the GA group and 109.57 ± 178.69 days in the LA group ($p < 0.01$). There was recurrence of pneumothorax in 43 of the 255 patients (16.86%); of the 43, 42 cases (97.7%) were from the GA group while one case (2.3%) was from the LA group. The rate of recurrence in the GA group was 18.7% and that of the LA group was 3.3%, which was significantly different ($p = 0.03$) (Table 3).

The mean operative time was 60.99 ± 22.57 minutes in the GA group and 50.57 ± 14.83 minutes in the LA group ($p = 0.02$). The total number of wedge resections performed was 1.93 ± 0.74 in the GA group and 1.48 ± 0.64 in the LA group ($p < 0.01$). The total length of lung resected was 7.50 ± 3.21 in the GA group and 5.85 ± 2.60 in the LA group ($p = 0.73$) (Table 4).

**Table 4.** Intraoperative details.

| Variable | GA Group | LA Group | *p*-Value |
|---|---|---|---|
| | (n = 225) | (n = 30) | |
| Operation time (minutes) | 60.99 ± 22.57 | 50.57 ± 14.83 | 0.02 |
| Total number of wedge resections | 1.93 ± 0.74 | 1.48 ± 0.64 | <0.01 |
| Total length of resected lung | 7.50 ± 3.21 | 5.85 ± 2.60 | <0.01 |
| Pleural coverage methods | | | 0.03 |
| Neoveil®with fibrin sealant | 205 | 29 | |
| Surgicel®with fibrin sealant | 20 | 1 | |

The correlation coefficients were not statistically significant between the rate of recurrence and the operative time (*p* = 0.52), the number of ports inserted (*p* = 0.74), the length of lung resected (*p* = 0.371), the technique used for pleural coverage (*p* = 0.65), the number of wedge resections performed (*p* = 0.26), the number of previous attacks (*p* = 0.17), or smoking history (*p* = 0.07; ex-smoker, *p* = 0.48; recent smoker, *p* = 0.07).

On the other hand, the difference in lung behavior according to the anesthetic technique used was analyzed from a biomechanical point of view (Figure 1) as follows.

- A/B in Figure 1 (lung expansion mechanism under general anesthesia): a double-lumen endotracheal tube is inserted into the airway and then the positive pressure is applied to the lungs. As a result, the lung volume becomes larger than the maximum expansion lung volume during breathing and more intra-alveolar air is retained. This phenomenon can generate high tension in the resection line and cause damage to the visceral pleura, which could precede the neogenesis of bullae.
- C/D in Figure 1 (lung expansion mechanism under local anesthesia): a catheter is inserted into the thorax and then negative pressure is applied to the lungs. As a result, the diaphragm elevates, resulting in less intra-thoracic negative pressure than in the procedure using a double-lumen endotracheal tube. This induces less tension in the resection line and causes less damage to the visceral pleura.

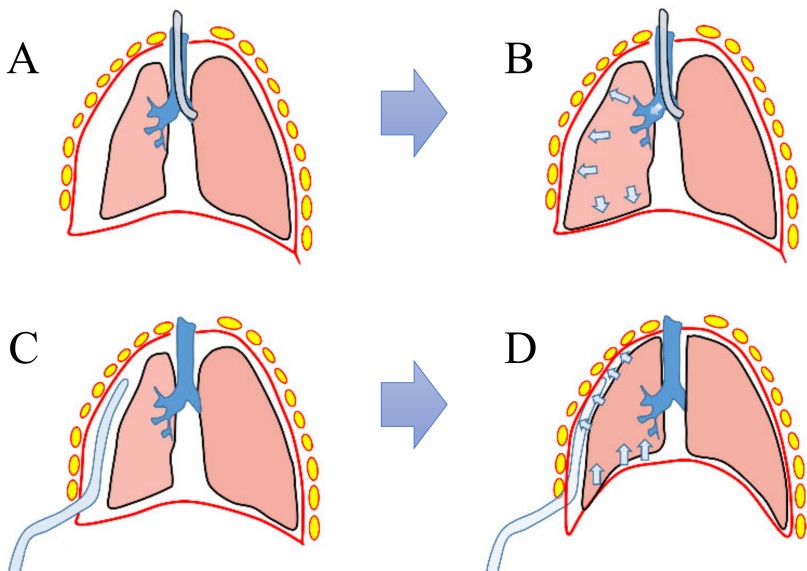

A/B: Lung expansion mechanism under general anesthesia
C/D: Lung expansion mechanism under local anesthesia

**Figure 1.** Diagram illustrating the biomechanical behavior of lung according to the anesthetic technique used.

## 4. Discussion

With the advancement of surgical techniques and equipment, VATS surgery can be easily adapted for the management of PSP. Despite the use of minimally invasive incisions, the field of anesthesia still aims to further minimize the damage to patients during a surgery. There are many studies comparing methods of anesthesia, specifically comparing GA with a double-lumen endotracheal tube with epidural anesthesia [2,11–13]. However, there have only been a few studies comparing postoperative clinical outcomes in patients with pneumothorax who underwent surgery under general and local anesthesia [10]. This study reviewed many variables to compare postoperative clinical outcomes in patients who underwent surgery for PSP under GA and LA at our center.

The surgical technique was more or less similar between the two groups: wedge resection of blebs was done with shallow shaving so as not to have excessive tension in the resection area, followed by visceral coverage using Neoveil®with glue. However, intraoperative management should be different for the two groups; in the case of GA, the exposed lung can be expanded by the anesthesiologist whenever required (if oxygen saturation decreases or if vital signs become unstable). However, with LA, the gradually collapsed lung cannot be expanded, even in a critical situation, unless the incision is sealed. Then, locking ports with valves at the entrance (usually used in laparoscopic surgery) need to be placed—which could make the lung expand due to the negative pressure through the ports at the end of the operation.

In the study reported here, we did not perform mechanical or chemical pleurodesis after resection, since mechanical pleurodesis could lead to bleeding and chest pain and chemical pleurodesis could cause acute lung injury due to pulmonary dissemination of talc particles. Recent studies have shown that pneumothorax generally occurs in visceral pleural lesions and that visceral strengthening should be aimed for to prevent recurrence, rather than adhesion between visceral and parietal pleura. Furthermore, mechanical or chemical pleurodesis was not preferable for reducing postoperative pain.

Performing thoracic surgery under LA can induce iatrogenic pneumothorax, resulting in desaturation, or, due to the manipulation of the lung parenchyma, activate the cough reflex—both of which increase the difficulty of the procedure by impeding easy manipulation. Therefore, unnecessary manipulation of the lung tissue should be minimized. It follows that planning ahead is important, using preoperative chest CT images to locate the blebs and plan the placement of ports.

The total duration of hospital stay was significantly shorter for the LA group than for the GA group—a result of the decreased time interval from admission to operation and operation to discharge. The interval between admission and operation was short because surgery under LA is the easiest way to arrange the schedule in our center. The interval between operation and discharge was significantly shorter for the LA group than for the GA group, which made a difference to the total duration of hospital stay. The duration of postoperative chest tube drainage especially was relatively shorter in the LA group than in the GA group and could be predicted from early independent ambulation and lung rehabilitation and early cessation of postoperative air leakage.

Reviewing the relationships between the clinical outcomes and the perioperative findings shows that the recurrence rate was significantly lower in cases operated upon with local anesthesia. There was no correlation between the various operative techniques, the operative time, or the length of resected lung parenchyma and the rate of recurrence. However, we hypothesize that some differences between the two groups during lung expansion had an impact on the rate of recurrence; this hypothesis is based on several articles that have shown that the neogenesis of bullae causes recurrences of the pneumothorax. Generally speaking, the neogenesis of bullae can be caused by the check valve mechanism affecting the staple line. In the GA group, the lung was usually expanded by positive pressure applied through the double-lumen endotracheal tube (Figure 1A,B); as the air entered the alveoli without sufficient removal, it could become an obstruction and cause initiation of neogenesis of bullae. However, in the LA group, a collapsed lung could be

expanded by negative pressure applied through the catheter in the thorax, which causes elevation of the diaphragm and decreases intrathoracic volume (Figure 1C,D).

The benefits of LA with sedation are as follows: less pain in the throat and decreased postoperative numbness. Under LA, there is no need to intubate with the double-lumen endotracheal tube. This eliminates the injuries associated with intubation; therefore, fewer patients complain of throat pain in LA groups than in GA groups [2,11–15]. During surgery under LA, the patient is in mild sedation. Owing to the risk of the patient waking up with pain, intraoperative procedures are performed carefully. This can lessen the external force on the intercostal space, which then decreases the acute postoperative pain.

The results of this study show that operations under LA might be feasible. However, there are some limitations: the two groups studied were not randomized, the LA group had a relatively short duration of follow-up (which might have led to the underestimation of the rate of recurrence), and the number of patients was small. The medical system in our center requests that patients with PSP are not fast-tracked; however, the Korean educational system is relatively tight and it may take a long time for students to be admitted, as mentioned in Section 2. These estrangements between the real world and medical system led us to study operations under LA, which do not require overly long hospital stays or preoperative periods. Until recently, postoperative outcomes after operations under LA were comparable with outcomes for operations under GA, as confirmed retrospectively by the present observational study. In the near future, we intend to initiate a randomized study to compare the postoperative outcomes.

Despite the limitations of this study and the difficulties of operating with LA, this study shows that surgery under LA with sedation might be both feasible and efficient in treating patients with PSP.

**Author Contributions:** Conceptualization, E.K. (Eunji Kim), C.-S.L. and H.Y.A.; investigation, E.K. (Eunji Kim) and H.Y.A.; formal analysis, E.K. (Eunji Kim) and H.Y.A.; resources, J.S.C., H.I., Y.D.K. E.K. (Eunsoo Kim) and H.Y.A.; project administration, C.-S.L.; writing—original draft preparation, E.K. (Eunji Kim), C.-S.L. and H.Y.A.; writing—review and editing, E.K. (Eunji Kim), C.-S.L. and H.Y.A. All authors have read and agreed to the published version of the manuscript.

**Funding:** This work was supported by the National Research Foundation of Korea (NRF) grant funded by the Korea Government (MSIT) (No. NRF-2020R1C1C1005004).

**Institutional Review Board Statement:** The study was conducted according to the guidelines of the Declaration of Helsinki and approved by the Institutional Review Board of Pusan National University Hospital (IRB No. 1908020084).

**Informed Consent Statement:** Informed consent was obtained from all subjects involved in the study.

**Data Availability Statement:** Not applicable.

**Conflicts of Interest:** The authors declare no conflict of interest. The funders had no role in the study design, data collection and analyses, writing of the manuscript, or in the decision to publish the results.

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
