# Peer review of "Postoperative Clinical Outcomes of Thoracoscopic Surgery under Local Anesthesia in Patients with Primary Spontaneous Pneumothorax"

_applsci, doi:10.3390/app11041468_

Round 1
Reviewer 1 Report
In this work, the Authors deal with the feasibility of VATS under LA compared to VATS under GA for the treatment of patients with recurrent PSP. Thoracic surgery under LA is one of the most debated topics, and this work adds another interesting experience in this field.
Here are my comments:
- The disproportion between the two groups is significative (225 vs 30); have you considered with your statistician some analysis to reduce this imbalance?
- If possible, for more completeness it would be better to add the mean weight and height as well as the smoking habit of the two groups in Table 1.
- Besides the wedge resection, why a pleurodesis technique to minimize the risk of recurrence was not considered? Please, provide an explanation in the Discussion.
- In line 197-201, you consider lung expansion as a possible cause of the higher number of recurrence in the GA group. Have you compared the smoking habit between the two groups? Smokers have more probability of recurrence. As previously stated, why a pleurodesis technique was not considered? Please, provide some explanation in the Discussion.
Author Response
Thank you for having me the chance of revision.
- The disproportion between the two groups is significative (225 vs 30); have you considered with your statistician some analysis to reduce this imbalance?- Not really. Because, the technique and surgical instruments have been developed in recent days. Just a few years ago, we couldn’t operate under LA. Until recent days, postoperative outcomes after operation under LA has been comparable with GA group, conservative study has performed retrospectively.(251~255)
- If possible, for more completeness it would be better to add the mean weight and height as well as the smoking habit of the two groups in Table 1.-I added in Table 2. Because I added a new table in 1.
- Besides the wedge resection, why a pleurodesis technique to minimize the risk of recurrence was not considered? Please, provide an explanation in the Discussion.
Thank you for a good point. I added “Especially in our series, we didn’t perform mechanical or chemical pleurodesis after resection, since mechanical pleurodesis could develop bleeding, chest pain, and chemical pleurodesis could cause acute lung injury due to pulmonary dissemination of talc particles. Based on recent studies, pneumothorax generally occurs in visceral pleural lesion and to prevent the recurrence, visceral strengthening should be followed, not by the adhesion between visceral and parietal pleura. Also for reducing the postoperative pain, mechanical or chemical pleurodesis was not preferable.” (204~209)
- In line 197-201, you consider lung expansion as a possible cause of the higher number of recurrence in the GA group. Have you compared the smoking habit between the two groups? Smokers have more probability of recurrence. As previously stated, why a pleurodesis technique was not considered? Please, provide some explanation in the Discussion.
Yes~!! I added this smoking history in table 2. But there was more percentage of smoker in LA group, which showed no significant difference between two groups.
Reviewer 2 Report
This manuscript examines the use of local anesthesia (LA) versus general anesthesia (GA) in the surgical management of spontaneous pneumothorax. The conclusions are that LA has substantial benefits over GA, for a number of reasons, the most important of which in my opinion are that the recurrence rate appears to be substantially lower and the period of hospitalisation is shorter.
However, these conclusions must be viewed with substantial caveats. These caveats are:
- The participants were not randomised, so it is impossible to exclude randomisation bias. Put simply, the LA patients may have had less severe disease, hence the outcomes would be expected to be better in this group.
- Although the authors state in the abstract that the "surgical technique" was relatively identical, in fact the method of lung re-inflation was completely different (which the authors explain and correctly comment on), namely positive intratracheal pressure under GA, versus pleural suction under LA, thus potentially ameliorating neogenesis of bullae. This point should receive more emphasis.
- At a more general level, my view is that what this manuscript shows is that in a select group of patients (not a randomised group) LA can be performed, and is likely to result in better outcomes. However, both the small size of the study and the failure to delineate selection criteria more specifically makes it difficult to judge who this sub-group should be.
In summary, my opinion is that this pilot study has shown valuable results and is consequently worthy of publication, but the conclusions would be substantially more powerful if the study were appropriately randomised, or the selection criteria for LA were clearly stated.
Minor points:
- Six criteria for surgical intervention were listed but no breakdown of these criteria were included in the demographics.
- One of the criteria for surgical intervention was "upcoming important examinations", and thus is not a medical indication (although I suppose a psychosocial argument could be made in relation to impact on mental health).
- This appears to be a retrospective study and should be identified as such.
- Line 179: "clogged" could be replaced with "sealed"
Author Response
Thank you for having me the chance of revision
- The participants were not randomised, so it is impossible to exclude randomisation bias. Put simply, the LA patients may have had less severe disease, hence the outcomes would be expected to be better in this group.--> Even in this retrspective conservative study, all patients were suffering from primary pneumothorax, so there was no severe disease in LA group and I added the table 1 which showed the prevalence of indication in each groups.
- Although the authors state in the abstract that the "surgical technique" was relatively identical, in fact the method of lung re-inflation was completely different (which the authors explain and correctly comment on), namely positive intratracheal pressure under GA, versus pleural suction under LA, thus potentially ameliorating neogenesis of bullae. This point should receive more emphasis.--> Yes~!! Thank you for good comments.So I added Generally, the neogensis of bullae might be caused by the check valve mechanism affecting the staple line. In the GA group, the lung was usually expanded by positive pressure applied through the double lumen endotracheal tube (Fig. 1A, B); as the air comes into the alveoli without sufficient removal, it could attribute an obstructing air and cause initiation of neogenesis of bullae(230~233)
- At a more general level, my view is that what this manuscript shows is that in a select group of patients (not a randomised group) LA can be performed, and is likely to result in better outcomes. However, both the small size of the study and the failure to delineate selection criteria more specifically makes it difficult to judge who this sub-group should be.-->Yes. You are right~!! However, The medical system in our center request not too fast track for patient with PSP, and Korean educational system is relatively tight and hard for students not to be admitted too long. These estrangement between real world and medical system made us study operation under LA, which needed not too much hospital stays and not too long preoperative days. Until recent days, postoperative outcomes after operation under LA has been comparable with GA group, conservative study has performed retrospectively. In near future, the randomized study has to be started to compare the postoperative outcomes. (251~257)
In summary, my opinion is that this pilot study has shown valuable results and is consequently worthy of publication, but the conclusions would be substantially more powerful if the study were appropriately randomised, or the selection criteria for LA were clearly stated.
Minor points:
- Six criteria for surgical intervention were listed but no breakdown of these criteria were included in the demographics.--> I added "table 1"
- One of the criteria for surgical intervention was "upcoming important examinations", and thus is not a medical indication (although I suppose a psychosocial argument could be made in relation to impact on mental health).-->I totally agree with you, but, in different situation in Korea. In Korea, Korean educationalou are system is relatively tight and hard, which could make the students under pressure, especially in exam days and try to avoid the reccurence of pneumothorax (Table 1).
- This appears to be a retrospective study and should be identified as such.-->Until recent days, postoperative outcomes after operation under LA has been comparable with GA group, conservative study has performed retrospectively. In near future, the randomized study has to be started to compare the postoperative outcomes. (253~257)
- Line 179: "clogged" could be replaced with "sealed"--> I changed. Thank you
This manuscript is a resubmission of an earlier submission. The following is a list of the peer review reports and author responses from that submission.
Round 1
Reviewer 1 Report
I congratulate the Authors for choosing this subject of so called "awaken surgery". To support their choice, the Authors cite possible complications connected to general anesthesia. Primary spontaneous pneumothorax is a disease generally affecting young, otherwise healthy, male patients. Therefore general aneshtesia complications in this category are almost absent
The Authors attribute a shorter hospitalization of patients operated in LA to an early mobilization, but, again, young patients are used to early walking and rehabilitation.
Why LA is easier to arrange in your Department?
An important finding could be the lowest relapse rate in patients undergoing LA surgery that Authors reported. But, as the Authors state, the shorter follow up might explain the gap between the two groups.
Author Response
Thank you for having the chance to revise my article.
For first question, I agree with your opinion.
As in your opinion, young patients don't have critical postoperative complications, however, as describing in the introduction, postoperative sore throat or nausea could happen, usually. And some patients in out patient clinics complain about the postoperative pain in throat. Comparing with GA group, postoperative immediate pain and pain in out patient clinic in LA group was relatively less than GA group. Especially, in young patients, they don't endure postoperative pain during admission. So, critical complications wouldn't be happened, but, minor complications such as pain, nausea could happen.
For second question, usually in LA group, the patients might mobilize as soon as they are awaken from slight sedation. They don't usually feel much pain even after the operation, because of long affects from injected local anesthesia. That's why the patients easily walk and eat.
For third, in our hospital, to request the general anesthesia, we need at least 2 days before operation. Especially if there comes the weekend, they need to wait at least 4 days. In our education system, more than 4 days absent might affect the patient distressed. In case with indications to be operated, we try to perform the operation as early as possible.
For forth, even though follow-up duration was longer in GA group, the recurrence was usually occurred in 3 months after operation. 109.57 days might not be short period to compare the recurrence rate.
Reviewer 2 Report
This article deals with an interesting topic: "the post-operative clinical outcome of VATS in patients with PSP".
This article seems quite organized. In my opinion, this original article deserve mention in the scientific literature. However, I would suggest that the authors consider the following points for the final version.
Abstract
I would suggest writing it better to improve the flow and readability of the text. In addition, abbreviations have to be defined at first use (see the acronym VAS)
Introduction
OK
Methods
Statistical analysis
What test did you use to assess normality? This is an important information to justify your choice to present the results as mean (± standard deviation) or median (interquartile range) and use parametric (or non-parametric) tests. In fact, when the data are normally distributed, descriptive statistic should be reported as meanstandard deviation and parametric test should be used. When the data are not normally distributed, descriptive statistics should be reported as median and range (minimum and maximum) or interquartile range (IQR).
In addition, please specify with what data and to do what the T-test and the chi-squared were used
Results
Lines 109-112. It is a repetition, this infomration has already been provided in the Methods section.Therefore, the authors must choose whether to include these data in the Methods or Result Section
line 118. There is an error. 20 or 29 patients in the LA group. Please correct
Line121: instead of writing...(p=0.11, 0.31, 0.21). Better to put a single value (p ≥ 0.11). The same thing at the line 126.
A similar problem at line 128...the authors write (p=0.00). In my opinion it would be better to put (p<0.01). The same thing at lines 130 and 133.
Lines 128, 133 and 137. Another problem, the authors talk about Table 2 and Table 3. But I don't see them.
Line 138: the authors talk about correlation coefficient (this test must also be included in the statistical analysis subsection).
Lines 140-141: see above; better to put a single value (p ≥ 0.17). To express all values, the authors could add a new table.
Figure 1; it should be included after its citation in the text
Discussion
OK
References
Please redo in agreement with the “Applied Sciences Instructions for Authors”.
Author Response
Thank you for having us to have revisioin
- abstract: I changed the abbreviations to full name. and is there any way to improve the flow and readability of the text~?
- since the number of data was over 30, the normal distribution was presumed. T-test was used in all the datas except, gender and the chi-squared was used in gender.
- All things were corrected as you recommended.
Round 2
Reviewer 1 Report
I thank the Authors for trying to ameloriate their paper. I believe that the premise behind this work is wrong. As already stated in the first version, I do not believe that this is the category of patients that can benefit most from local anesthesia surgery.
Reviewer 2 Report
1) With regard the abstract, mine was just a suggestion. If you like it in its current form, okay. After all, it's your article.
2) For normal distribution, your response is not appropriate. For small sample (from 5 to 50) the normal distribution should be tested with one of the following tests: Shapiro–Wilk test or Kolmogorov–Smirnov test
3) Last and penultimate row of Table 2 (i.e. Recurrence and Follow-up duration (days)); I would be better writing P values as <0.01 instead of 0.00
4) Table 3, fourth row (i.e. Pleural coverage methods): I would be better writing P value as 0.03 instead of 0.029
5) References should be formatted according to the Journal guidelines